# Natural Activators of Autophagy Reduce Oxidative Stress and Muscle Injury Biomarkers in Endurance Athletes: A Pilot Study

**DOI:** 10.3390/nu15020459

**Published:** 2023-01-16

**Authors:** Alessandra D’Amico, Chiara Fossati, Fabio Pigozzi, Paolo Borrione, Mariangela Peruzzi, Simona Bartimoccia, Filippo Saba, Annachiara Pingitore, Giuseppe Biondi-Zoccai, Luigi Petramala, Fabrizio De Grandis, Daniele Vecchio, Luca D’Ambrosio, Sonia Schiavon, Luigi Sciarra, Cristina Nocella, Elena Cavarretta

**Affiliations:** 1Department of Movement, Human and Health Sciences, University of Rome “Foro Italico”, 00135 Rome, Italy; 2Villa Stuart Sport Clinic, FIFA Medical Centre of Excellence, 00135 Rome, Italy; 3Department of Clinical Internal, Anesthesiology and Cardiovascular Sciences, Sapienza University of Rome, 00161 Rome, Italy; 4Mediterranea Cardiocentro, 80122 Naples, Italy; 5Fondazione Santa Lucia IRCCS, 00179 Roma, Italy; 6Department of General and Specialistic Surgery “Paride Stefanini”, Sapienza University of Rome, 00161 Rome, Italy; 7Department of Medical-Surgical Sciences and Biotechnologies, Sapienza University of Rome, 04100 Latina, Italy; 8Department of Translational and Precision Medicine, “Sapienza” University of Rome, 00185 Rome, Italy; 9Department of Clinical Medicine, Public Health, Life and Environmental Sciences, University of L’Aquila, 67100 Coppito, Italy

**Keywords:** muscle injury, oxidative stress, autophagy, trehalose, polyphenols, athletes

## Abstract

Background: Oxidative stress and impaired autophagy are directly and indirectly implicated in exercise-mediated muscle injury. Trehalose, spermidine, nicotinamide, and polyphenols possess pro-autophagic and antioxidant properties, and could therefore reduce exercise-induced damage to skeletal muscle. The aim of this study was to investigate whether a mixture of these compounds was able to improve muscle injury biomarkers in endurance athletes through the modulation of oxidative stress and autophagic machinery. Methods and Results: sNOX2-dp; H_2_O_2_ production; H_2_O_2_ breakdown activity (HBA); ATG5 and p62 levels, both markers of autophagic process; and muscle injury biomarkers were evaluated in five endurance athletes who were allocated in a crossover design study to daily administration of 10.5 g of an experimental mixture or no treatment, with evaluations conducted at baseline and after 30 days of mixture consumption. Compared to baseline, the mixture intake led to a remarkable reduction of oxidative stress and positively modulated autophagy. Finally, after the 30-day supplementation period, a significant decrease in muscle injury biomarkers was found. Conclusion: Supplementation with this mixture positively affected redox state and autophagy and improved muscle injury biomarkers in athletes, allowing for better muscle recovery. Moreover, it is speculated that this mixture could also benefit patients suffering from muscle injuries, such as cancer or cardiovascular patients, or elderly subjects.

## 1. Introduction

Exhaustive exercise, as well as novel or unaccustomed intensive exercise, can result in muscle injury, leading to various undesirable consequences such as negative impacts on muscle function, and interference with daily physical activities may occur. According to the most accepted theory, high mechanical tension and metabolic changes, particularly due to a high intracellular calcium concentration with a loss of cellular homeostasis, are involved in muscle injury [1]. Moreover, the magnitude of the damage is influenced by the mode, intensity, and duration of the exercise, as well as by the training status of participants [2]. Since contracting skeletal muscles leads to the production of reactive oxygen species (ROS) that are potentially harmful to tissues, the impact of oxidative stress on skeletal muscle function as an effector of exercise-induced muscle damage has now become a source of great scientific interest. An overabundance of free radical generation in the body that exceeds its scavenging rate during physical exercise potentially predisposes individuals to detrimental effects on skeletal muscles and highlights the biological implications of this relationship [3]. Among the primary intracellular sources of ROS during exercise, NADPH oxidase (NOX) family proteins can produce more superoxide anions than those produced by other sources such as mitochondria or xanthine oxidase. Specifically, NOX2 is the main source of cytosolic ROS in skeletal muscle during moderate-intensity exercise, as demonstrated in p47phox-mutated mice harboring a loss-of-function mutation in the regulatory NOX2 subunit, p47phox. In these mice, completely abolished ROS production was observed after an acute moderate-intensity treadmill exercise [4]. Moreover, our group demonstrated that elite athletes who practiced highly intensive sports displayed higher levels of NOX2, paralleled by a significant reduction in antioxidant capacity activation, compared to amateur athletes [5]. These data support the role of NOX2 as a major source of ROS generation during exercise.

In the homeostasis of redox status, the autophagic process plays a key role in removing oxidized cellular components and in regulating cellular ROS levels. In general, autophagy is a major intracellular catabolic process that plays an essential role in the survival and homeostasis of cells [6,7]. Autophagy can be induced by various stimuli such as starvation, endoplasmic reticulum (ER) stress, hypoxia, mitochondrial dysfunction, and oxidative stress [8]. Under these stress conditions, autophagy-related genes (ATGs) are activated for the formation, growth, and maturation of the autophagosome. In particular, ATG5 is an important protein in the context of autophagy as it plays a critical role in the formation of the autophagosome, a sequestering vesicle with a double membrane that fuses with the lysosome for the degradation of cytoplasmic material [9]. p62/SQSTM1 is a ubiquitin-binding scaffold protein that interacts with ubiquitinated cargo proteins, recruiting them into the growing autophagosome membrane for degradation. p62 is degraded following the initiation of autophagy; thus, p62 accumulation when autophagy is inhibited may be used as a marker of reduced autophagic flux [10].

The activation of autophagic flux mediates the clearance of protein aggregates, damaged intracellular organelles, and ROS. Autophagy further triggers transcription factor activation and degrades impaired organelles and proteins to eliminate excessive ROS in cells. On the other hand, ROS can modulate autophagic activity through transcriptional and post-translational mechanisms. Therefore, autophagy may play an antioxidant role in protecting cells from oxidative stress, and autophagic impairments are closely linked to different physiological and pathological processes. Specifically, in muscle cells, impaired autophagy results in a pathological cascade that may also contribute to cell damage. Consequently, positive modulation of autophagic machinery to reduce ROS might be an appropriate strategy to counteract the harmful consequences of muscle injury.

Natural activators of autophagy were reported to exert beneficial effects in preclinical models of cardiovascular diseases, acting on specific molecular targets such as the mechanistic target of rapamycin complex 1 (mTORC1), transcription factor EB (TFEB), and 5′ adenosine monophosphate-activated protein kinase [11,12]. Among several autophagy-inducing agents, trehalose is a nonreducing disaccharide, comprising an *α*, *α*-1,1-glucosidic bond between two *α*-glucose units. As a pharmacological agent, trehalose shows several advantages including high hydrophilicity, chemical stability, and strong resistance to acid hydrolysis and cleavage by glucosidases owing to its nonreducing property [13]. Trehalose has received great attention for its beneficial cardiac effects, as suggested by the improvement of left ventricular function and cardiac remodeling [14] and the activation of lipophagy in the mouse heart [15].

Spermidine, polyphenols, and nicotinamide are also potent natural activators of autophagy, as indicated by accumulating lines of evidence.

Spermidine is a naturally occurring endogenous polyamine with a wide range of beneficial effects, including cardiovascular protection, immune system regulation, and neuroprotective effects [16,17,18]. For example, a higher intake of dietary spermidine is associated with a decreased risk of cardiovascular disease and all-cause mortality [19]. In mice, spermidine has been reported to significantly attenuate cardiac dysfunction [20] and to reduce necrotic core formation and lipid accumulation in the atherosclerotic plaque [21] via induction of autophagy [20,21]. Nicotinamide was also reported to regulate autophagy, accelerating autophagic degradation of mitochondria in human cells and restoring autophagy in stroke-prone spontaneously hypertensive rats [22,23]. In a previous study, the combination of trehalose with a mixture of spermidine, nicotinamide, catechin, and epicatechin reduced platelet activation and oxidative stress in platelets isolated from smokers or patients with atrial fibrillation or metabolic syndrome. Moreover, the mixture increased the production of nitric oxide, angiogenesis, and cell viability in HUVEC [24]. As trehalose, spermidine, nicotinamide, and polyphenols could reduce the impact of several ROS-related risk factors [12,24], the purpose of this study was to evaluate whether the administration of a mixture of natural activators of autophagy was able to reduce muscle injury biomarkers in athletes through downregulation of ROS production mediated by improvement of autophagic function and oxidative stress.

## 2. Materials and Methods

### 2.1. Study Design

This was an independent, nonrandomized, crossover study. A schematic representation of the study design is provided in Figure 1.

All athletes provided written informed consent before enrolment. All athletes were allocated to a 30-day treatment sequence with 10.5 g of the mixture (twice a day), and then a 30-day washout was established before allocation to the no-treatment condition in a crossover design. The same conditions of athletic training and competition were maintained for the duration of the 3-month study. The study protocol was approved by the local ethical board of Sapienza University of Rome (ref. 5382/19) and was conducted according to the principles of the 1975 Declaration of Helsinki. No funding was received directly or indirectly from the manufacturer or supplier.

### 2.2. Participants

To evaluate the effect of a mixture of trehalose, spermidine, nicotinamide, and polyphenols, a crossover interventional study was performed in five endurance athletes (two males, three females). Athletes were enrolled in the study after the preparticipation screening, performed before the beginning of the official competitive season. One athlete was excluded before enrolment because of a recent traumatic musculoskeletal injury. Specifically, two athletes practiced rowing, two long-distance running, and one cycling. During the training and competitive season, athletes were engaged in a mean training time of 11.8h per week (including warm-up, aerobic training reaching 75% of the maximal heart rate, strength training, and cool down).

Inclusion criteria were as follows: endurance athletes aged >18 years to 50 years, of either sex, who practiced intense specific endurance training sessions of more than 1 h per session and more than three sessions per week for a total of at least 8 h/week (>6 METs), and had at least 3 years of competitive experience.

Exclusion criteria were as follows: the occurrence of musculoskeletal injuries in the last month before enrolment, cardiovascular contraindication to competitive sport, pregnancy, presence of chronic disease, and consumption of drugs or supplements that could influence the experimental protocol.

Athletes with clinical conditions influencing oxidative stress or athletes who experienced injuries during the study were excluded. The treatment mixture was administered twice a day for 1 month.

### 2.3. Mixture Composition

The mixture was obtained from Princeps srl (Piasco, Cuneo, Italy). The mixture composition is reported in Table 1.

### 2.4. Blood Sampling and Preparations

All blood samples were collected in the morning (8–9 a.m.), from the antecubital vein of fasting athletes in a seated position. Blood samples were collected prior to the beginning of the training season for the elite athletes, at the same time as the clinical evaluation, to better normalize the ROS production [25]. The training in the days before blood sampling was free for athletes and all athletes abstained from training for 24h before blood sampling, as requested by our study protocol. The training sessions were not standardized for athletes. Plasma and serum samples were collected in BD Vacutainers (Franklin Lakes, NJ, USA) with or without anticoagulant (trisodium citrate, 3.8%, 1/10 (*v*/*v*)), respectively. The blood was centrifuged at 300× *g* for 10 min at room temperature (RT). The supernatants were divided into aliquots and stored at −80 °C for analysis. Biomarkers of muscle injury, namely creatine kinase (CK), lactate dehydrogenase (LDH), and myoglobin levels; oxidative stress, assessed via blood levels of soluble NOX2-derivative peptide (sNOX2-dp), a marker of NOX2 activation; H_2_O_2_ production; and serum hydrogen peroxide (H_2_O_2_) breakdown activity (HBA) were evaluated. Moreover, markers of autophagy, namely ATG5 and p62, were assessed in plasma at baseline (T0) and after 1 month of treatment and after the washout period before and after the 30-day no-treatment period (T1M).

### 2.5. Evaluation of Oxidative Stress

Serum hydrogen peroxide (H_2_O_2_) breakdown activity (HBA) was measured using an HBA assay kit (Aurogene, code HPSA-50). The % HBA was calculated according to the following formula: % HBA = Ac − As/Ac × 100, where Ac is the absorbance of H_2_O_2_ (1.4 mg/mL) and As is the absorbance in the presence of the serum sample.

NOX2 activity was measured in serum as sNOX2-dp using a previously reported ELISA method [26]. Values were expressed as pg/mL; intra- and interassay coefficients of variation (CV) were 8.95% and 9.01%, respectively.

The hydrogen peroxide (H_2_O_2_) levels were measured using a colorimetric assay as previously described [27]. A standard curve of H_2_O_2_ (0–200 μM) was constructed for each assay. Briefly, 50 μL of cell supernatant was mixed with 3,3′,5,5′ tetramethylbenzidine (50 μL, Sigma-Aldrich, St. Louis, MO, USA) in 0.42 mol/L citrate buffer, pH 3.8, and 10 μL of horseradish peroxidase (52.5 U/mL, Sigma-Aldrich, St. Louis, MI, USA). The samples were incubated at room temperature for 20 min, and the reaction was stopped by the addition of 10 μL 18 N sulfuric acid. The reaction product was measured spectrophotometrically at 450 nm and expressed as μM.

### 2.6. Evaluation of Muscle Damage

Muscle damage was evaluated via serum creatine kinase (CK), lactate dehydrogenase (LDH), and myoglobin levels analyzed using a commercial ELISA kit (Antibodies-online GmbH, Aachen, Germany; EIAab Wuhan, China; DRG Instruments GmbH, Marburg, Germany).

### 2.7. Evaluation of Autophagy Biomarkers

#### 2.7.1. Plasma ATG5 Detection

Quantitative determination of ATG5 in plasma samples was performed using the sandwich enzyme immunoassay technique (MyBioSource, No. MBS2602759). The sample concentration was determined using a microplate reader set to 450 nm and values were expressed as ng/mL. Intra- and interassay coefficients of variation were ≤8% and ≤12%, respectively.

#### 2.7.2. Plasma p62 Detection

Plasmatic p62 was analyzed using a commercial ELISA kit (FineTest, No. EH10842). The protein concentration was determined using a microplate reader set to 450 nm and values were expressed as ng/mL. Intra- and interassay coefficients of variation were <8 and <10%, respectively.

### 2.8. Sample Size Calculation

A provisional sample size computation was performed assuming no effect in the control arm and a decrease from 450 to 400 IU/L in serum levels of CK in the experimental arm, assuming an error variance of 1000 and a correlation of 0.75. Five subjects were needed within a repeated-measure design aiming for 0.80 power and two-tailed 0.05 alpha.

### 2.9. Statistical Analyses

Continuous variables were reported as mean ± standard deviation according to normal distribution. The assessment of treatment effects was on repeated-measure ANOVA. Bivariate analysis was performed using Pearson and Spearman correlation tests. A value of *p* < 0.05 was considered statistically significant. All analyses were performed using GraphPad Software-Prism 7.

## 3. Results

Demographic and clinical characteristics of the athletes are reported in Table 2. The study was carried out during the competitive season over three consecutive months (30-day treatment, 30-day washout, 30-day no-treatment period).

Athletes reported no adverse effects related to the mixture consumption and remained asymptomatic during the whole study period. No injuries or trauma were reported.

### 3.1. Muscle Damage

After 1 month, a significant difference between treatments (mixture vs. no treatment) was found with respect to biomarkers of muscle damage such as CK, LDH, and myoglobin levels (*p* < 0.05; *p* < 0.01, Figure 2a–c).

Compared with the baseline, a significant decrease was observed in CK levels after mixture intake (from 463.1 ± 102.2 to 322.4 ± 93.9 mU/mL, *p* < 0.01) (Figure 2a and Table 3). LDH and myoglobin also showed decreases after mixture intake (from 120.1 ± 40.0 to 72.3 ± 20.3, mU/mL, *p* < 0.05, and from 114.9 ± 27.1 to 97.4 ± 19.5 ng/mL, *p* < 0.05, respectively) (Figure 2b,c and Table 3). No changes were observed after the no-treatment period for any of the measured parameters (from 483.3 ± 103.4 to 484.3 ± 86.7 mU/mL for CK; from 119 ± 29.8 to 121.9 ± 29.2 mU/mL for LDH; from 111.8 ± 25.4 to 112.9 ± 23.5 ng/mL for myoglobin) (Figure 2a–c and Table 3).

### 3.2. Oxidative Stress

After 1 month, significant differences between treatments (mixture vs. no treatment) were found with respect to sNOX2-dp release (*p* < 0.01; Figure 3a), H_2_O_2_ (*p* < 0.05; Figure 3b), and HBA (*p* < 0.01; Figure 3c).

The pairwise comparisons showed that sNOX2-dp and H_2_O_2_ significantly decreased after mixture intake (from 17.1 ± 5.9 to 8.6 ± 2.3 pg/mL, *p* < 0.01, and from 17.8 ± 2.6 to 10.5 ± 2.3 μM, *p* < 0.05, respectively), while no changes were observed after the no-treatment period (from 17.9 ± 3.8 to 18.8 ± 3.3 pg/mL, and from 18.3 ± 2.8 to 19.4 ± 4.9 μM, respectively) (Figure 3a,b and Table 3).

Conversely, HBA, an index of the antioxidant power and a measure of the percentage of H_2_O_2_ removed from serum via activation of antioxidant systems [28], significantly increased after mixture intake (from 42.1 ± 7.2 to 52.1 ± 4.0%, *p* < 0.05). No changes were observed after the no-treatment period (from 38.3 ± 6.6 to 37.7 ± 6.6%) (Figure 3c and Table 3).

### 3.3. Autophagy

Autophagy was evaluated via the plasmatic levels of P62 and ATG5. This analysis revealed that mixture administration significantly increased the levels of ATG5 protein (*p* < 0.05; Figure 4a) and decreased serum p62 levels (*p* < 0.01; Figure 4b).

Compared with the baseline, serum levels of ATG5 significantly increased (from 108 ± 23.7 to 157.9 ± 33.5 ng/mL, *p* < 0.05) and p62 significantly decreased (from 90.9 ± 13.9 to 71.8 ± 16.6 ng/mL, *p* < 0.01), while no changes were observed in the same athletes after the no-treatment period (from 104.5 ± 30.6 to 107.6± 37.4 ng/mL for ATG5 and from 93.7± 6.1 to 93.2 ± 9.5 for p62) (Figure 4a,b and Table 3).

### 3.4. Linear Correlation

A linear correlation analysis showed that ∆ (expressed as the difference between values before and after mixture intake) CK correlated with ∆ p62 (R^2^ = 0.42; *p* = 0.04) (Figure 5a), ∆ H_2_O_2_ (R^2^ = 0.55; *p* = 0.01) (Figure 5b), ∆ sNOX2-dp (R^2^ = 0.58; *p* = 0.01) (Figure 5c), and ∆ HBA (R^2^ = 0.51; *p* = 0.02) (Figure 5d). Moreover, ∆ LDH correlated with ∆ sNOX2-dp (R^2^ = 0.52; *p* = 0.02) (Figure 5e) and with ∆ HBA (R^2^ = 0.41; *p* = 0.04) (Figure 5f).

## 4. Discussion

This study explored the impact of oral supplementation with a mixture of natural activators of autophagy and antioxidants on muscle biomarkers in athletes. The novel finding of the present study is the improvement of muscle damage biomarkers in endurance athletes after a 30-day period of treatment with a new mixture. This improvement was accompanied by reductions in NOX2 activation and ROS production, indicative of reduced oxidative damage, and changes in autophagy-related proteins, indicative of restoration of autophagic function.

Supplementation with antioxidants may be a viable strategy to mitigate exercise-induced maladaptive changes. Much evidence supports the beneficial effects of acute and chronic supplementation with fruit-derived polyphenols, which occur via a mechanism most likely to be related to antioxidant and vascular effects [29]. Among polyphenol-rich nutrients, cocoa beans are rich in polyphenols, especially catechin and epicatechin. A previous study demonstrated that polyphenol-rich nutrient supplementation with dark chocolate positively modulated redox status and reduced exercise-induced muscular injury biomarkers in elite football athletes [30]. Moreover, in vitro studies support the positive effect of cocoa-derived polyphenols, as skeletal muscle myoblast cell lines treated with polyphenol extracts displayed decreased oxidative stress and muscle injury biomarker levels [5].

Exercise is an external stimulus of oxidative stress that causes physiological adaptative changes in the skeletal muscle. Both the structure and function of skeletal muscle cells can be modified by exercise-induced oxidative stress. In general, during moderate exercise, ROS are produced at relatively low rates in skeletal muscle fibers; these levels are closely related to positive effects on gene expression, regulation of cell signaling, and modulation of contractile force. In contrast, strenuous or acute exercise produces high levels of ROS that can cause muscle fatigue and dysfunction as a result of damage to cellular components such as proteins and organelles. The alteration of redox hemostasis impacts exercise capacity and could be harmful to health. The roles of ROS and oxidative stress in the regulation of skeletal muscle function have been extensively demonstrated [3]. Furthermore, oxidative stress, assessed via NOX2 activation and H_2_O_2_ production, was significantly higher in elite football players compared to active control subjects who practiced moderate-to-intense physical activity. These changes were coincident with increased levels of biomarkers of muscle injury such as LDH, CK, and myoglobin [30]. Moreover, elite football players compared to amateurs displayed increased concentrations of dopamine, which plays a key role in favoring platelet activation and in promoting NOX2-mediated oxidative stress, ultimately favoring muscle injury [5].

In this crossover study, changes in the muscle injury parameters were found after mixture intake, as demonstrated by the significant decreases in CK, LDH, and myoglobin levels. Moreover, after a 30-day intervention with the mixture, athletes displayed reduced levels of oxidative stress biomarkers, namely NOX2dp and H_2_O_2_. Restoration of a balanced redox status was also supported by a significant increase in the percentage of HBA, as an index of improved ROS detoxifying mechanisms, confirming the beneficial effect of this mixture on oxidative status. Conversely, endurance athletes that did not receive treatment with the mixture showed the same high levels of oxidative stress and muscle injury biomarkers after 30 days compared to baseline.

Besides oxidative stress, another cellular process that regulates muscle cell homeostasis is autophagy. Autophagy is a major intracellular homeostatic process that clears damaged organelles, cytoplasmic components, protein aggregates, long-lived proteins, and damaged organelles through the autophagosome–lysosome system. Recently, autophagy has gained attention for its role in metabolic homeostasis and skeletal muscle disease progression. Indeed, skeletal muscle is one of the tissues with the highest basal autophagy flux and also with the greatest capacity to increase autophagy flux [31]. Exercise is an external factor that can commonly induce alterations in autophagic flux; however, how exercise intensity modulates autophagy in skeletal muscle remains unclear, as different responses of autophagy have been reported following different types of physical exercise [32]. For example, in mice, acute exercise has been shown to induce autophagy in skeletal muscle, as autophagic proteins such as LC3-II and p62 were found to be significantly increased [33]. However, acute exercise has also been associated with decreased expressions of LC3-II, Beclin-1, Atg7, and LAMP-2, consistent with attenuated autophagic signaling [34]. The same contradictory results in autophagic response were reported after long-term chronic exercise [32].

Both oxidative stress and autophagy response exert ambivalent effects, as they can be detrimental or beneficial according to their delicate balance. Indeed, redox and autophagy signaling are also interrelated, even if little is known about the precise mechanisms of ROS-regulated autophagy. In this regard, ROS has been shown to both promote and impair autophagy in skeletal muscle [35]. The unresolved question about ROS crosstalk with autophagic signaling opens up prospects for studies to define and discover new and effective therapeutic strategies to modulate these processes.

In this regard, autophagy activators such as trehalose can be used in the management of several conditions in which autophagy plays a beneficial role. Recent studies have demonstrated that the ingestion of a trehalose solution, as a source of carbohydrates, aids in the maintenance of performance in the later stages of prolonged exercise compared to glucose solution [36,37,38]. However, studies that have examined its properties in athletes are still limited.

The present results showed that administration of the mixture increased ATG5 and decreased p62 levels, thus restoring autophagy. These changes are consistent with previous findings demonstrating that trehalose, in combination with polyphenols, reduced NOX2 activation and oxidative stress and restored autophagy in endothelial cells and platelets [24]. No changes were observed in endurance athletes not treated with the mixture.

The study had limitations and implications. First, the sample size was too small to evaluate the differences between the two groups. In fact, this study should be considered a proof-of-concept study that provides potentially useful evidence to further understanding of the mechanisms of muscle injury; future studies in larger populations are warranted to confirm these results. Second, a clinical evaluation of muscle damage was not performed. The muscle biomarkers analyzed can be considered parameters that indicate a muscle injury. Third, in this study, a known product with components able to improve the autophagic process and oxidative stress was used. However, the hypothesis that other products with the same characteristics may also have similar results cannot be excluded.

Finally, according to these results, it can be speculated that the administration of this mixture could also benefit patients suffering from muscle injuries, such as cancer or cardiovascular patients, or it could be administered to elderly subjects. Indeed, it is known that aging-related muscle atrophy is the most common type of muscle atrophy in humans and is associated with significant impairment of function, such as slowing of movement and muscle weakness.

## 5. Conclusions

Overall, the results suggest that chronic administration of a mixture of trehalose, spermidine, nicotinamide, and polyphenols mitigated the detrimental effects associated with increased oxidative stress and impaired autophagy, which ultimately led to reduced levels of muscle injury biomarkers in endurance athletes.

Confirmation of the effect in a larger population is needed to assess whether this mixture may represent a new intervention strategy to provide health benefits to endurance athletes.

## Figures and Tables

**Figure 1 nutrients-15-00459-f001:**
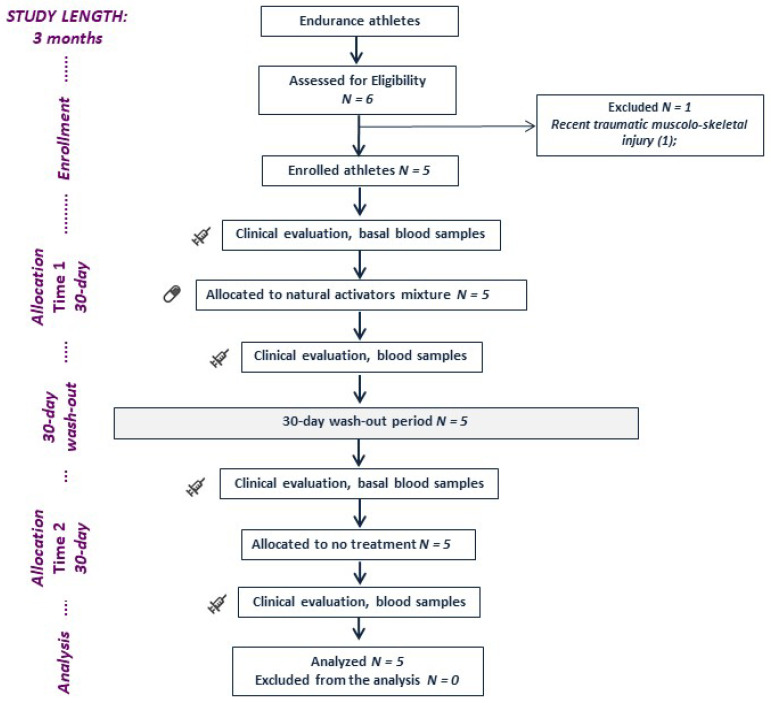
A schematic representation of the study design.

**Figure 2 nutrients-15-00459-f002:**
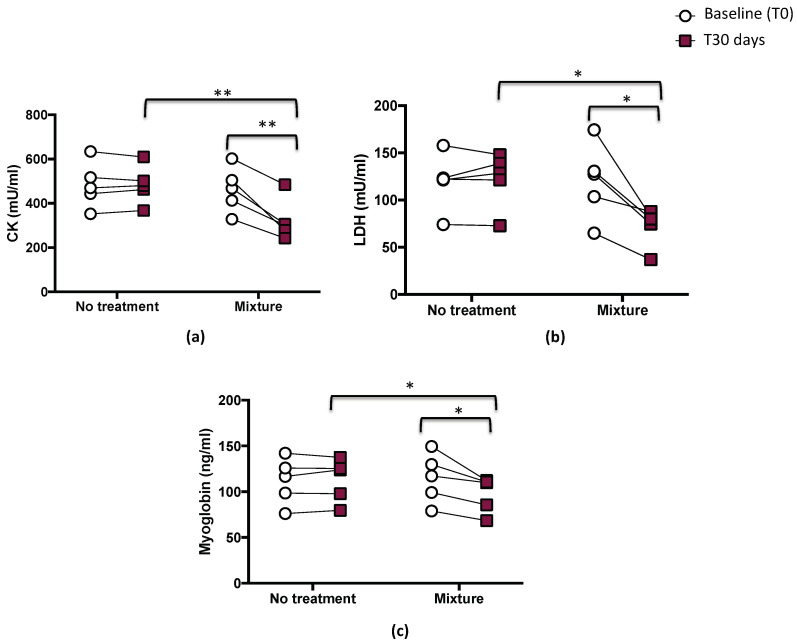
Serum levels of CK (**a**), LDH (**b**), and myoglobin (**c**) before (T0) and 1 month (T30 days) after mixture intake (*n* = 5) or no-treatment period (*n* = 5) in athletes. Data are expressed as mean ± SD. * *p* < 0.05, ** *p* < 0.01.

**Figure 3 nutrients-15-00459-f003:**
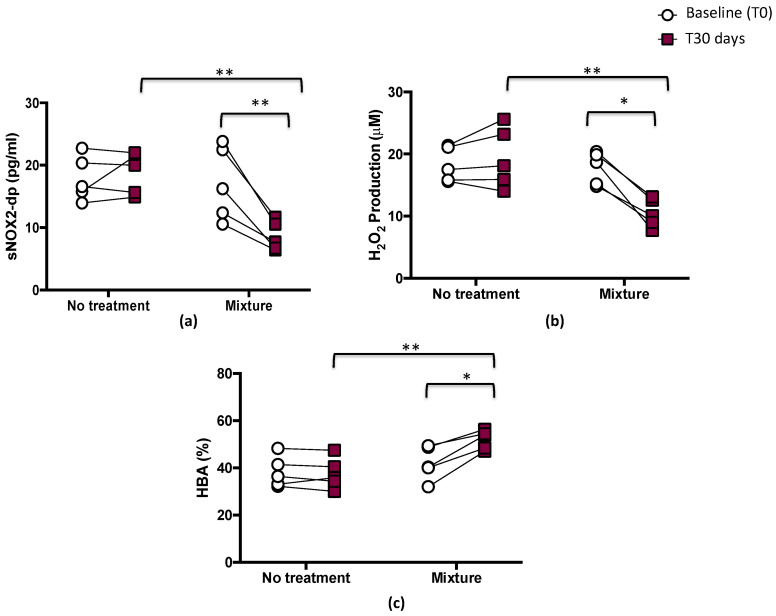
Serum soluble NOX2-derived peptide (sNOX2-dp) (**a**), serum H_2_O_2_ (**b**), and blood HBA (**c**) before (T0) and 1 month (T30 days) after mixture intake (*n* = 5) or no-treatment period (*n* = 5) in athletes. Data are expressed as mean ± SD. * *p* < 0.05, ** *p* < 0.01.

**Figure 4 nutrients-15-00459-f004:**
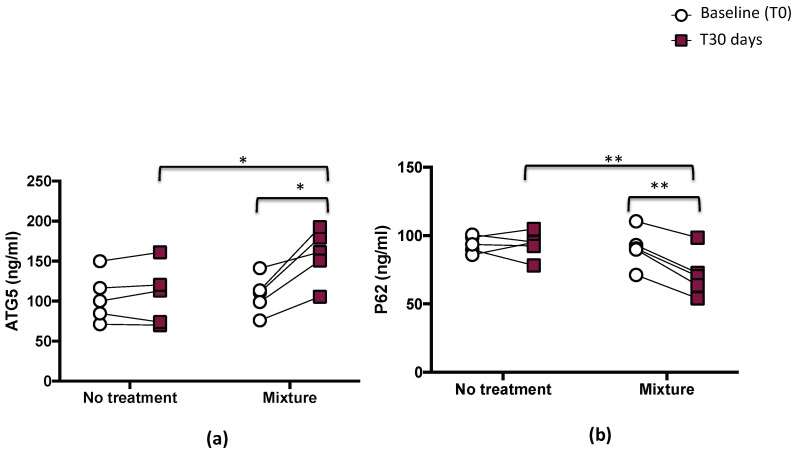
Serum ATG5 (**a**) and p62 (**b**) before (T0) and 1 month (T30 days) after mixture intake (*n* = 5) or no-treatment period (*n* = 5) in athletes. Data are expressed as mean ± SD. * *p* < 0.05, ** *p* < 0.01.

**Figure 5 nutrients-15-00459-f005:**
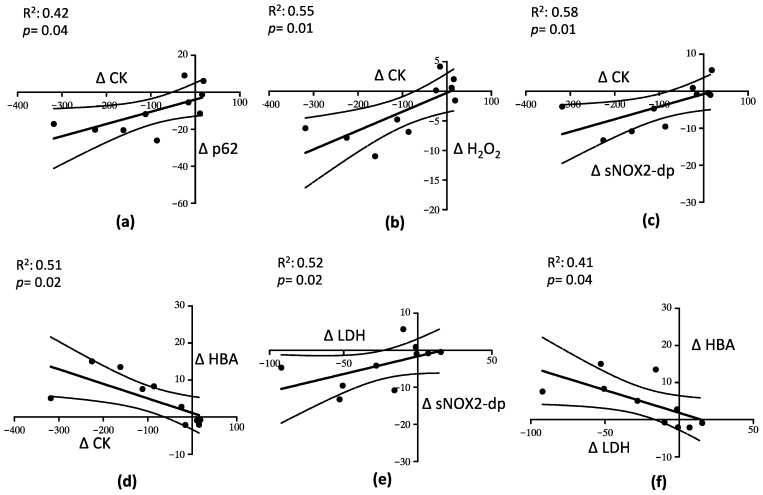
Linear correlation analysis between the parameters describing muscle injury, oxidative stress and autophagy. Relationship between Δ CK and Δ p62 (**a**), Δ CK and Δ H_2_O_2_ (**b**), Δ CK and Δ sNOX2-dp (**c**), Δ CK and Δ HBA (**d**), Δ LDH and Δ sNO2-dp (**e**), and Δ LDH and Δ HBA (**f**).

**Table 1 nutrients-15-00459-t001:** Mixture composition.

Mixture Composition	Grams (g)
Trehalose	7.5
Spermidine	0.00225
Camellia Sinensis e.s.	0.075
Catechins	0.0375
Vitamin C	0.06
Niacin	0.00375
Silica	0.225
Microcrystalline cellulose	0.075
Orange aroma	0.3

**Table 2 nutrients-15-00459-t002:** Characteristics of athletes.

	Athletes (*n* = 5)	Lower/Upper 95% CI of Mean
Age (years)	21.8 ± 1.3	20.2–23.4
Sex (M/F)	2/3	/
Cholesterol (mg/dL)	162.4 ± 27.9	127.8/197.0
BMI	20.6 ± 2.5	17.5/23.7
Glycemia (mg/dL)	83.6 ± 6.7	75.2/91.9
Training per week (h)	11.8 ± 3.5	8.0/15.5
Sport practice (years)	12.7 ± 4.6	8.2/17
Systolic blood pressure (mmHg)	112.8 ± 4	107/117
Diastolic blood pressure (mmHg)	72 ± 7	65/78
Resting heart rate (bpm)	54 ± 5	48/60
Peak heart rate (bpm)	168 ± 12	163–178
Maximum workload (METs)	12.5 ± 2	10.5–14.3

**Table 3 nutrients-15-00459-t003:** Biomarker levels before and after treatment and no-treatment period.

	Before Mean ± SD	After Mean ± SD	*p*-Value	Lower/Upper 95% CI of Mean before	Lower/Upper 95% CI of Mean after
CK (mU/mL)					
No treatment	483.1 ± 103.4	484.3 ± 86.7	0.99	354.9/611.7	376.5/592.0
Mixture treatment	463.1 ± 102.2	322.4 ± 93.9	** *p* < 0.01	336.2/590.0	205.8/439.1
LDH (mU/mL)					
No treatment	119.7 ± 29.8	121.9 ± 29.3	0.99	82.7/156.8	85.5/158.2
Mixture treatment	120.1 ± 40.0	72.3 ± 20.3	* *p* < 0.05	70.42/169.8	47.1/97.5
Myoglobin (ng/mL)					
No treatment	111.8 ± 25.4	112.9 ± 23.6	0.99	80.3/143.4	83.6/142.1
Mixture treatment	114.9 ± 27.1	99.4 ± 21.6	* *p* < 0.05	81.9/148.6	72.5/126.2
sNOX2-dp (pg/mL)					
No treatment	17.9 ± 3.5	18.8 ± 3.3	0.99	13.5/22.3	14.6/22.8
Mixture treatment	17.1 ± 5.9	8.6 ± 2.4	** *p* < 0.01	9.7/24.4	5.6/11.6
H_2_O_2_ (μM)					
No treatment	18.3 ± 2.8	19.3 ± 4.8	0.99	14.7/21.8	13.3/25.4
Mixture treatment	17.8 ± 2.6	10.5 ± 2.3	* *p* < 0.05	14.5/21.1	7.6/13.3
HBA (%)					
No treatment	38.3 ± 6.6	37.7 ± 6.6	0.99	30.0/46.5	29.4/45.9
Mixture treatment	42.1 ± 7.2	52.2 ± 4.1	* *p* < 0.05	33.2/51.1	47.0/57.1
ATG5 (ng/mL)					
No treatment	194.5 ± 30.5	107.6 ± 37.4	0.99	66.5/142.4	61.1/154.0
Mixture treatment	108.0 ± 23.7	155.9 ± 37.5	* *p* < 0.05	78.5/137.5	109.4/202.4
P62 (ng/mL)					
No treatment	93.7 ± 6.1	93.1 ± 9.6	0.99	86.1/101.3	81.2/105.1
Mixture treatment	90.9 ± 13.9	71.8 ± 16.6	** *p* < 0.01	73.7/108.3	51.3/92.5

**Legend:** ATG5 = autophagy-related genes 5; CK = creatine kinase; LDH = lactate dehydrogenase; H_2_O_2_ = hydrogen peroxide; HBA = hydrogen peroxide breakdown activity; sNOX2-dp = soluble NOX2-derived peptide. * *p* < 0.05, ** *p* < 0.01.

## Data Availability

The data presented in this study are available on request from the corresponding author.

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
