# Peer review of "Natural Activators of Autophagy Reduce Oxidative Stress and Muscle Injury Biomarkers in Endurance Athletes: A Pilot Study"

_nutrients, 2023, doi:10.3390/nu15020459_

Round 1

Reviewer 1 Report

Thank you for the opportunity to review this manuscript, which investigate whether a mixture of trehalose, spermidine, nicotinamide, and polyphenols were able to improve muscle injury biomarkers in endurance athletes through the modulation of oxidative stress and autophagic machinery. At face value, it appears that the study design was well-thought out and accurately replicates what could conceivably be implemented in practice. In addition, the paper mentioned an important topic as the oxidative stress and impaired autophagy are, directly and indirectly, implicated in exercise-mediated muscle injury. Trehalose, spermidine, nicotinamide, and polyphenols have shown to display pro-autophagic and antioxidant properties, eventually reducing exercise-induced damage of skeletal muscle. For this reason, it has a valuable and socially useful content. The manuscript is very well written and structured. I have some suggestions for the authors.

1.     For future investigations should be indicated at the end of the abstract as well as in the conclusions section some directions about the results in relation to the topic

2.     The most recent updates should be mentioned in the Introduction.

3.     The introduction doesn’t explain the basic concepts used in the paper. Some of these are presented in the methodology section, in terms of how they were measured, however it would have been necessary to clarify their general content in the introduction section.

Methods

4.     Add a section that clarified the study design. Point 2.1.

5.     Add a section with the participants 2.2. Add info about the group, the precedence of sample. Skills, etc…

6.     Improve the inclusion criteria

7.     Maybe include a table with the different biomarker, mean and acronym

8.     Can you add in the participants section a paragraph with the sample size calculated with G-power? Is a key factor. You only have 5 participants

9.     A schematic representation is necessary to elucidate the study.

Statistical analysis.

10.  The point 2.8 is not enough explained. More information about sample size is needed.

Results

11.  The figures is too small.

12.  If you add info about individual results the figures are valid. If no add the information, maybe you need ad a box plot with the individual data of each participant to understand the behaviour intra-subject and observe the trend. The Figure need to be improved, you have please be consistent.

13.  Tables need the same p-value format, and confidential interval 95% upper and lower could be added.  

14.  Why no add the figure of correlation with IC 95%?

15.   The values are incorrect. Only add two decimals in all text, please be consistent.

16.  Discussions is inappropriate, making reference to the results of other studies, but, the theoretical and practical implications of the research are vaguely mentioned. It necessary improve it. In addition, with only 5 participants, your discussion is powerful and challenging statements  

Author Response

Thank you for the opportunity to review this manuscript, which investigate whether a mixture of trehalose, spermidine, nicotinamide, and polyphenols were able to improve muscle injury biomarkers in endurance athletes through the modulation of oxidative stress and autophagic machinery. At face value, it appears that the study design was well-thought out and accurately replicates what could conceivably be implemented in practice. In addition, the paper mentioned an important topic as the oxidative stress and impaired autophagy are, directly and indirectly, implicated in exercise-mediated muscle injury. Trehalose, spermidine, nicotinamide, and polyphenols have shown to display pro-autophagic and antioxidant properties, eventually reducing exercise-induced damage of skeletal muscle. For this reason, it has a valuable and socially useful content. The manuscript is very well written and structured. I have some suggestions for the authors.

1. For future investigations should be indicated at the end of the abstract as well as in the conclusions section some directions about the results in relation to the topic

Answer: Amended as suggested (see page 1 lines 50-53 and page 13 lines 457-462)

2. The most recent updates should be mentioned in the Introduction.

Answer: In the Introduction section, we added some recent updates about the role of trehalose, and spermidine (ref 14-15 and 19-20; page 3 lines 113-115 and lines 120-123).

3. The introduction doesn’t explain the basic concepts used in the paper. Some of these are presented in the methodology section, in terms of how they were measured, however it would have been necessary to clarify their general content in the introduction section.

Answer: As suggested, we now added and clarified basic concepts in the introduction section (see page 2, lines 86-94)

4. Add a section that clarified the study design. Point 2.1.

Answer: Thank you for pointing out this important section. The study design has now been added as section 2.1 and all details of the study design have been further clarified. To add more clarity, the study design has been resumed in a new figure 1, based on CONSORT diagram. In particular, we stated: "This was an independent, non-randomized, cross-over study. A schematic representation of the study design is provided in Figure 1.

All athletes provided written informed consent before enrolment. All athletes were allocated to a 30-day treatment sequence with 10.5 g of mixture (twice a day), then 30-day of washout was established before allocation to no-treatment in a cross-over design. The same conditions of athletic training and competition have been maintained for the duration of the 3-month study. The study protocol was approved by the local ethical board of Sapienza-University of Rome (ref. 5382/19) and was conducted according to principles of the 1975 Declaration of Helsinki. No funding, directly or indirectly was received from the company manufacturer or supplier (see point 2.1 and Figure 1)

5. Add a section with the participants 2.2. Add info about the group, the precedence of sample. Skills, etc…

Answer: Thank you, a new 2.2 section dedicated to participants has been added to the manuscript. Demographic and clinical data related to the study population have now been provided in section 2.2 and in table 2 (see point 2.2 and Table 2).

6. Improve the inclusion criteria

Answer: Thank you, the inclusion criteria have been further detailed in section 2.2 Participants, specifically " Inclusion criteria were: endurance athletes aged > 18 years to 50 years, both sexes, who practice intense specific endurance training sessions of more than 1 hour per session and more than 3 sessions per week, for at least a total of 8h/week (>6 METs) and had at least 3 years of competitive experience." (see page 5 lines 175-178).

7. Maybe include a table with the different biomarker, mean and acronym

Answer: We added a Table (Now Table 3) with the different biomarkers analysed, mean value and acronyms.

8. Can you add in the participants section a paragraph with the sample size calculated with G-power? Is a key factor. You only have 5 participants

Answer: We have clarified in revised section 2.8 that ‘A provisional sample size computation was performed assuming no effect in the control arm and a decrease from 450 to 400 IU/L in serum levels of CK in the experimental arm, assuming an error variance of 1000 and a correlation of 0.75, 5 subjects were needed within a repeated-measure design aiming for 0.80 power and 2-tailed 0.05 alpha.’ (see page 6 lines 250-255).

9. A schematic representation is necessary to elucidate the study.

Answer: As suggested, we added a Graphical abstract.

Statistical analysis.

10. The point 2.8 is not enough explained. More information about sample size is needed.

Answer: Specifically, revised section 2.8 now explicitly states that ‘‘A provisional sample size computation was performed assuming no effect in the control arm and a decrease from 450 to 400 IU/L in serum levels of CK in the experimental arm, assuming an error variance of 1000 and a correlation of 0.75, 5 subjects were needed within a repeated-measure design aiming for 0.80 power and 2-tailed 0.05 alpha.’’ (see page 6 lines 250-255).

Results

11. The figures is too small.

Answer: We re-made the figures to improve the size.

12. If you add info about individual results the figures are valid. If no add the information, maybe you need ad a box plot with the individual data of each participant to understand the behaviour intra-subject and observe the trend. The Figure need to be improved, you have please be consistent.

Answer: According to your suggestion, we re-made the figures with individual data.

13. Tables need the same p-value format, and confidential interval 95% upper and lower could be added.

Answer: We added in Table 2 and new Table 3 confidential interval 95% upper and lower.  

14. Why no add the figure of correlation with IC 95%?

Answer: The figure of correlation with IC 95% has now been added (See New Figure 5)

15. The values are incorrect. Only add two decimals in all text, please be consistent.

Answer: Amended as suggested.

16. Discussions is inappropriate, making reference to the results of other studies, but, the theoretical and practical implications of the research are vaguely mentioned. It necessary improve it. In addition, with only 5 participants, your discussion is powerful and challenging statements

Answer: We modified the discussion by better integrating the results of this study with respect to the literature evidence and adding the practical implications of the research (see Discussion section).

Reviewer 2 Report

Trehalose, spermidine, nicotinamide, and polyphenols have been shown to display pro-autophagic and antioxidant properties, eventually reducing exercise-induced damage to skeletal muscle. To investigate whether a mixture of trehalose, spermidine, nicotinamide, and polyphenols were able to improve muscle injury biomarkers in endurance athletes through the modulation of oxidative stress and autophagic machinery. Supplementation with a mixture of trehalose, spermidine, nicotinamide, and polyphenols positively affected redox state 44 and autophagy and improved muscle injury biomarkers in athletes with a mechanism possibly related to NOX2-mediated oxidative stress downregulation and autophagic flux upregulation.

Overall, this is a very interesting and straightforward observational pilot study. However, there are several questions that need to be addressed before further consideration.

1.       Why CK, LDH, and myoglobin levels were not increased in the no treatments group (Fig.1)? How this is in fact a muscle injury model? In all three figures, the authors should show individual data points, instead of the Mean with SD.

2.       Again, unlike blood HBA level, why were serum H2O2 and Serum soluble NOX2-derived peptide levels not increased without treatments (fig. 2)? This raised questions about the author’s hypothesis.

3.       More justification is required for enhanced Serum ATG5 and suppressed p62 levels with the treatments as well as why these levels did not change without treatments.

4.       The limitations section needs to be elaborated, including very small samples.

5.       The time of data collection should be included as these parameters should fluctuate throughout the day.

6.       This is mostly observational studies and mechanistic detail is limited, therefore, it will be very useful to summarize overall findings using a model.

7.       The mixture of trehalose, spermidine, nicotinamide, and polyphenols has been bought for a biotech company; however, the authors need to provide more justification for using this mixture.

8.       The muscle injury model is questionable as several parameters used in this study did not change over time, particularly during extensive exercise without the treatment. The authors need to provide why at the basal level these parameters are high in the serum and why they did not change over time.

9.       The authors should at least include some discussion about muscle injury examples- such as aging, obesity, cancer, or other disease conditions and make an assessment if the mixture of trehalose, spermidine, nicotinamide, and polyphenols will be helpful for other muscle injury models.  

Author Response

Trehalose, spermidine, nicotinamide, and polyphenols have been shown to display pro-autophagic and antioxidant properties, eventually reducing exercise-induced damage to skeletal muscle. To investigate whether a mixture of trehalose, spermidine, nicotinamide, and polyphenols were able to improve muscle injury biomarkers in endurance athletes through the modulation of oxidative stress and autophagic machinery. Supplementation with a mixture of trehalose, spermidine, nicotinamide, and polyphenols positively affected redox state 44 and autophagy and improved muscle injury biomarkers in athletes with a mechanism possibly related to NOX2-mediated oxidative stress downregulation and autophagic flux upregulation.

Overall, this is a very interesting and straightforward observational pilot study. However, there are several questions that need to be addressed before further consideration.

1. Why CK, LDH, and myoglobin levels were not increased in the no treatments group (Fig.1)? How this is in fact a muscle injury model?

Answer: We found no changes in the levels of biomarkers of muscle injury because the same conditions of training and competition have been maintained for the duration of the study. Endurance sport practice is a model of muscle injury because it has been demonstrated that long-term physical training of endurance athletes results in increased steroid biosynthesis, fatty acid metabolism, oxidative stress, and biomarkers of muscle injury (PMID: 29305667). In fact, the levels of muscle injury biomarkers remain high in the no-treatment group compared to the treatment group.

In all three figures, the authors should show individual data points, instead of the Mean with SD.

Answer: As suggested, we now reported individual data points in all Figures.

2. Again, unlike blood HBA level, why were serum H2O2 and Serum soluble NOX2-derived peptide levels not increased without treatments (fig. 2)? This raised questions about the author’s hypothesis.

Answer: As above reported, endurance sports result in increased oxidative stress and decreased antioxidant status. Therefore, our hypothesis was to verify if the treatment with antioxidant and pro-autophagic substances was able to ameliorate oxidative stress and thus the muscle injury.

3. More justification is required for enhanced Serum ATG5 and suppressed p62 levels with the treatments as well as why these levels did not change without treatments.

Answer: For the same reasons, endurance athletes show an impairment of the autophagic process due to an increase in the inflammatory process that does not change in the no-treatments group because the same conditions of training and competition have been maintained for the duration of the study.

4. The limitations section needs to be elaborated, including very small samples.

Answer: Amended as suggested (see page 13 lines 447-456).

5. The time of data collection should be included as these parameters should fluctuate throughout the day.

Answer: Thank you for pointing out this information. We already stated in section 2.4 Blood samples and preparation that "All blood samples were collected in the morning (8–9 a.m.), from the antecubital vein in seated position in fasting athletes." We moved up to the beginning of this section further detail on the blood collection and added that the clinical evaluation of athletes was performed immediately prior to blood sampling.

6. This is mostly observational studies and mechanistic detail is limited; therefore, it will be very useful to summarize overall findings using a model.

Answer: We added a Graphical Abstract to describe mechanistic detail and main findings

7. The mixture of trehalose, spermidine, nicotinamide, and polyphenols has been bought for a biotech company; however, the authors need to provide more justification for using this mixture.

Answer: We used these products in the mixture because they are able to improve the autophagic process (see trehalose and spermidine, as demonstrated by data in the literature: PMID: 29724354, PMID: 28408448) and reduce oxidative stress (see polyphenols, as demonstrated by data in the literature: PMID: 35237163) simultaneously. We cannot exclude that other products with the same characteristics may also have similar results. We now added this issue in the limitations (see page 13 lines 453-456).

8. The muscle injury model is questionable as several parameters used in this study did not change over time, particularly during extensive exercise without the treatment. The authors need to provide why at the basal level these parameters are high in the serum and why they did not change over time.

Answer: See answer point 1

9. The authors should at least include some discussion about muscle injury examples- such as aging, obesity, cancer, or other disease conditions and make an assessment if the mixture of trehalose, spermidine, nicotinamide, and polyphenols will be helpful for other muscle injury models.

Answer: Thank you for this important question. Now we reported this issue in the conclusions section (see page 13 lines 457-462).

Round 2

Reviewer 1 Report

Great job. The work was improved significantly.

Author Response

We thank the reviewer for his/her comments.

Reviewer 2 Report

The authors did not put any serious effort into addressing questions 1, 2, 3, and 8 raised in the previous version. The muscle injury model is inappropriate and I am not sure when we are using treatment.  They have to be addressed appropriately. 

The author has indicated they have included graphical abstract, however, I was not able to see that in the revised version. 

Author Response

The authors did not put any serious effort into addressing questions 1, 2, 3, and 8 raised in the previous version. The muscle injury model is inappropriate and I am not sure when we are using treatment. 

They have to be addressed appropriately. 

Answer: We apologize to the reviewer if we have not answered your previous questions satisfactorily.

We used endurance athletes as a muscle injury model because, according to several studies (doi:10.2165/00007256-198603050-00006, doi: 10.1017/S0007114508926544, doi: 10.1136/bjsm.2003.006502) strenuous endurance exercise induces muscle damage and impairs muscle function. Moreover, high-volume endurance training is associated with an accumulation of chronic skeletal muscle damage, predisposing these athletes to a greater risk of impaired performance, training intolerance, and associated-exercise chronic fatigue (doi: 10.1136/bjsm.2003.006502).

The author has indicated they have included graphical abstract, however, I was not able to see that in the revised version. 

Answer: We have now re-uploaded the Graphical Abstract.